# MiR-581/SMAD7 Axis Contributes to Colorectal Cancer Metastasis: A Bioinformatic and Experimental Validation-Based Study

**DOI:** 10.3390/ijms21186499

**Published:** 2020-09-05

**Authors:** Xiaojuan Zhao, Shuzhen Liu, Bianbian Yan, Jin Yang, Erfei Chen

**Affiliations:** 1Key Laboratory of Resource Biology and Biotechnology in Western China, Ministry of Education, School of Life Sciences, Northwest University, Xi’an 710069, China; xjzhao1992@163.com (X.Z.); lsz15709483140@163.com (S.L.); bbyan2018@163.com (B.Y.); yangjin@nwu.edu.cn (J.Y.); 2Institute of Preventive Genomic Medicine, School of Life Sciences, Northwest University, Xi’an 710069, China

**Keywords:** colorectal cancer, microRNA, bioinformatics, signaling pathway, metastasis

## Abstract

Metastasis is a well-known poor prognostic factor and primary cause of mortality in patients with colorectal cancer (CRC). Recently, with the progress of high through-put sequencing, aberrantly expressed non-coding RNAs (ncRNAs) were found to participate in the initiation and development of cancer. However, the mechanisms of ncRNA-mediated regulation of metastasis in CRC remain largely unknown. In this study, we systematically analyzed the expression network of microRNAs (miRNAs) and genes in CRC metastasis using bioinformatics, and discovered that the miR-581/SMAD7 axis could be a potential factor that drives CRC metastasis. A dual luciferase report assay and protein analysis confirmed the binding relationship between miR-581 and SMAD7. Further functional assays revealed that miR-581 inhibition could suppress cell proliferation and induce apoptosis in SW480 cells. Up-regulation or down-regulation of miR-581 could both affect cell invasion capacity and modulate epithelial to mesenchymal transition (EMT) via a SMAD7/TGFβ signaling pathway. In conclusion, our findings elucidated that miR-581/SMAD7 could be essential for CRC metastasis, and may serve as a potential therapeutic target for CRC patients.

## 1. Introduction

Colorectal cancer (CRC) is a common type of cancer with a relatively high incidence and morbidity worldwide [1]. Metastasis is one of the primary causes of CRC deaths, and patients with metastasis have a poor survival rate of 10% within 5 years [2]. Studies have revealed that inactive tumor suppressors (*APC*, *DCC*) act as driver genes for CRC initiation [3]. However, CRC metastasis is a multi-step process with various gene and epigenetic alterations, and the underlying molecular mechanisms are still elusive. An in-depth study and identification of novel targets for CRC metastasis could be of benefit for basic research and clinical application.

In recent years, microRNA (miRNA), a 19–25 nucleotide (nt) endogenous small RNA, has been widely studied. The abnormal expression of miRNAs has been found to be involved in tumor progression [4,5]. Through binding to the 3′- untranslated region (3′-UTR) of target genes, miRNAs regulate gene expression level and further affect downstream signaling pathways [6]. However, systematic analysis of miRNA/gene regulation during the process of CRC metastasis is still largely unexplored.

In the present study, we performed Short Time series Expression Miner (STEM) [7], a software application designed for the analysis of short time series expression datasets, to identify miRNAs that enriched in the same expression pattern. In numerous clinical studies, STEM was applied to time series miRNA expression datasets to discover the miRNAs that have same expression pattern [8]. Meanwhile, we performed weighted gene correlation network analysis (WGCNA) [9] to reveal the correlation of genes and to search for significantly correlated gene modules with CRC metastasis. By performing enrichment analysis in this gene module, we identified that the TGFβ signaling pathway was observably enriched. Finally, we found that up-regulation of miR-581 may act as a novel oncogene and target SMAD Family Member 7 (SMAD7). Several studies revealed miR-581 dysregulation in tumor samples. However, the function and regulation network of miR-581 is still unclear. Previous studies reported Smad7/TGFβ-mediated metastasis in various types of tumors [10,11,12], including colorectal cancer [13], indicating that miR-581 could be a novel regulator of metastasis. Here, we aimed to explore the role of miR-581 in CRC and elucidate the underlying mechanism involved in the process of metastasis. The study will also help us understand the epigenetic regulation of SMAD7 during CRC development.

## 2. Results

### 2.1. STEM Clustering on MiRNA Expression

To explore the difference in miRNA expression in CRC samples, we pre-processed miRNA data (Appendix A) and performed STEM tool, and found that two profiles were statistically significant within the eight model profiles (Figure 1A). MiRNAs in profiles 6 and 7 are summarized in Appendix A. We focused on the miRNAs in pattern 7 that displayed a continuous up-regulation pattern (Figure 1B). Among these miRNAs, miR-581, miR-922, miR-1206, miR-1263, miR-1278, and miR-1289 have not been reported in studies on the function of metastasis in cancers.

MiRDB, TargetScan, and microRNA.org were then applied to predict the potential target genes which may participate in CRC metastasis. Detailed overlapping potential target gene lists of above miRNAs are shown in Appendix A.

### 2.2. Identification of CRC Metastasis-Related Gene Modules and Pathways 

In order to discover the core genes in CRC metastasis, we pre-processed mRNA data of samples and performed weighted gene correlation network analysis (WGCNA) on mRNA expression profiles (Appendix A). We set the soft power for network construction at 10 (Appendix A), with minimum module size 30, and a total of 40 modules were excavated (Figure 2A). To find the module that displayed the most significant correlation with CRC metastasis trait, we calculated the module-trait correlation coefficients, and the results showed that the turquoise module had the higher correlation coefficient (Figure 2B). Subsequently, to discover the underlying biological pathways behind CRC metastasis trait, we performed Kyoto Encyclopedia of Genes and Genomes(KEGG) enrichment analysis on the top genes of the turquoise module (Figure 2C). The enrichment results are displayed in Appendix A.

### 2.3. Conjoint Analysis of MiRNA and mRNA Expression Profile in CRC.

KEGG analysis results show metastasis-related biological pathways in CRC: PI3K-Akt, TGF-beta, mTOR, Jak-STAT. The results confirm that the genes in turquoise module indeed correlate with CRC metastasis. 

In order to investigate which miRNA regulates the metastasis-related pathways through targeting genes that function as crucial mediators in pathways, we performed conjoint analysis between miRNA target genes and metastasis-related pathways. We discovered that miR-581 could bind to 3′-UTR of SMAD7 and mTOR, the critical regulation genes of TGF-beta and mTOR signaling pathways, respectively. Similarly, SMAD7 is also the overlapping target of miR-1263. Meanwhile, no crucial genes of these pathways appear in the target gene lists of the other miRNAs.

In the miRanda database, we observed that the conservation between miR-581 and SMAD7 (0.7231) or MTOR (0.7373) was higher than that between miR-1263 and SMAD7 (0.5984). The mirSVR score (indicating binding capacity) was higher in miR-581/SMAD7 (−1.3238) as compared with miR-581/MTOR (−0.8403) and miR-1263/SMAD7 (−0.3757). Theoretically, these reflect that miR-581 could more efficiently suppress SMAD7. Together with the published data that confirms that SMAD7 functions in metastasis, we focused on the miR-581/SMAD7 axis in the CRC model.

### 2.4. MiR-581 Tightly Binds to 3′-UTR of SMAD7

With the use of online prediction software microRNA.org, we found that miR-581 could tightly bind to the 3′-UTR of SMAD7 (Figure 3A). To confirm the physical interaction between miR-581 and SMAD7, we performed a dual-luciferase reporter assay and found that miR-581 could significantly repress the reporter activity (wild-type 3′-UTR of SMAD7) in SW480 cells (Figure 3B, *p* < 0.05), while having no effect on that of the mutant-type. Further expression analysis showed that SMAD7, mRNA, and protein level changed after miR-581 mimics or inhibitor transfection (Figure 3C,D). Particularly, miR-581 negatively correlated with SMAD7 in advanced patients with metastasis in The Cancer Genome Atlas (TCGA) cohort (Appendix A). These results demonstrate that miR-581 directly binds to SMAD7 and regulates its expression.

### 2.5. Decreased Level of MiR-581 Inhibited Cell Proliferation and Induced Apoptosis

The biological process in which miR-581 participates has not been well studied. In this study, for the first time, we assessed the role of miR-581 in CRC phenotype. The CCK-8 assay showed that miR-581 overexpression had no effect on SW480 cell growth rate. Meanwhile, knockdown of miR-581 inhibited cell proliferation as compared with negative control (inhibitor NC) or the blank group at the 48-h checkpoint (Figure 4A). Similar results were also found by flow cytometry detection. Cell apoptosis rate was measured 48 h after transfection, as was inhibition of miR-581induced apoptosis in SW480 cells (Figure 4B). 

### 2.6. MiR-581 Promotes Cell Invasion Capacity and Modulates Epithelial to Mesenchymal Process

To explore the role of miR-581 in CRC cell invasion, a transwell assay was performed in SW480 transfected cells. As shown in Figure 5A, the invasion capacity in miR-581 mimics group was significantly enhanced compared with the blank or NC group. Conversely, miR-581 inhibition notably weakened the invasion capacity of SW480 cells (Figure 5B, *p* < 0.01).

Previous studies have demonstrated that the SMAD7/TGFβ axis plays an essential role in epithelial to mesenchymal transition (EMT). Accordingly, in this study, we hypothesized that miR-581 could promote EMT in CRC by targeting SMAD7. Firstly, we observed that overexpression of miR-581 could encourage epithelial SW480 cells to adopt a spindle-like phenotype (Figure 6A). Western blot assays show that overexpression of miR-581 downregulates E-cadherin expression while upregulating N-cadherin and Vimentin levels (Figure 6B). Conversely, inhibition of miR-581 shows the opposite effects (Figure 6C). These results suggest that miR-581 could regulate CRC invasion and induce cells to undergo EMT in vitro.

### 2.7. MiR-581 Induced Malignant Phenotype of CRC via SMAD7/TGFβ Pathway

Since SMAD7 was a direct target of miR-581, we then carried out rescue experiments to confirm that SMAD7 was required for the miR-581-associated phenotypes in CRC cells. As shown in Figure 7A–D, SMAD7 knockdown significantly enhanced cell growth, invasion, and EMT capacity. After co-transfection of SMAD7-specific siRNA and miR-581 inhibitor, these phenotypes weakened as compared with the SMAD7 knockdown group. The above results suggest that miR-581 inhibition could partially reverse the promoting effect of SMAD7-knockdown on the regulation of CRC cells, and further confirm that miR-581 could promote cell invasion and EMT via SMAD7/TGFβ pathway.

## 3. Discussion

Abnormal expression of miRNAs in different stages has different effects on tumor initiation and progression of CRC. A wide range of miRNAs have been found in tumors, and healthy tissues which are associated with patient’s diagnosis and prognosis, such as miR-143, down-regulated in different stages of CRC, have been reported to be associated with colon cancer progression, prognosis, and metastasis [14,15]. During the process from benign adenoma to malignant carcinoma in CRC patients, miRNA molecules involved in key signaling pathways will be alternated [16]. For instance, in the progression of colon cancer, miR-135 and miR-34 are involved in the Wnt signaling pathway, and loss of miR-126 leads to amplification of PI3K signal in the PI3K-AKT signaling pathway [16,17,18]. However, how the non-coding part of the genome is involved in CRC metastasis remains largely unexplored.

Considering the dynamic changes in microRNA expression during CRC tumorigenesis and metastasis, we applied the STEM algorithm to identify significant profiles (profiles 6 and 7, *p* < 0.05). The significantly upregulated profile 7 was further characterized and investigated. Among 89 microRNAs, six novel molecules were selected for investigation into their function as diagnostic and/or therapeutic targets for CRC, including miR-581 miR-922, miR-1206, miR-1263, miR-1278 and miR-1289. At the same time, this study used the WGCNA algorithm for the purpose of identifying clinical feature-related gene co-expression modules. A large number of studies have been published using the WGCNA algorithm to screen clinical feature-related gene co-expression networks and biomarkers in several types of tumors [19,20]. We identified that the turquoise module had the highest correlation with metastasis trait. The functional enrichment analysis of the turquoise module that was correlated with the metastasis trait indicated that the genes in this module were enriched in tumor metastasis-associated KEGG pathways (PI3K-Akt, TGF-beta, mTOR, Jak-STAT, etc.). By comprehensively analyzing the microRNAs′ potential target genes and tumor pathways′ core genes, together with the conservation and mirSVR score of miRNAs and its targets, we ultimately identified miR-581 and SMAD7 of TGF-beta signal pathways.

The function and mechanism of miR-581 in cancer has not been reported. However, the results of several studies suggest that the dysregulation of miR-581 is involved in the development of cancer. For example, miR-581-associated single-nucleotide polymorphisms located on the MUC4 3′UTR may reduce the interaction between miR-581 and MUC4 mRNA, promote the expression of oncoprotein MUC4, and be associated with the incidence of gastric cancer [21]. Additionally, miR-581 is down-regulated in patients with hepatocellular carcinoma (HCC) and can be used as a novel marker for tumors and chronic liver disease [22]. 

SMAD7, a member of SMAD family, is reported to be associated with the development of a variety of cancers. Recent studies have shown that SMAD7 is present in common malignancies and acts as a tumor suppressor, including lung cancer, liver cancer, gastric cancer, bladder cancer, triple negative breast cancer, and osteosarcoma [11,23,24,25,26,27,28]. In CRC, reduced level of SMAD7 is found to be involved in the process of cancer growth as well as the process of EMT and invasion [29]. Studies also show that SMAD7 gene polymorphisms are associated with CRC risk in patients with Lynch syndrome [30]. The miR-581/SMAD7 axis we investigated here could also be a supplement to SMAD7 epigenetic modification. A variety of onco-miRNAs were discovered targeting SMAD7 [10,23,31,32,33,34]. Since the miRNA-target network is complex and not a ‘one to one’ model, the core miRNA that regulates SMAD7 level still needs to be further explored.

As we predicted, miR-581 could promote CRC cell invasion via the SMAD7/TGFβ pathway. Furthermore, our results reflect that miR-581 inhibition could also suppress CRC cell proliferation and induce apoptosis. Previous study has suggested SMAD7 can induce growth and inhibit apoptosis in colon adenocarcinoma cells and acts as a tumor promoter [35], indicating that miR-581-mediated apoptosis might be SMAD7/TGFβ-pathway-independent. Moreover, in the results of miR-581 target genes, TNFSF10 [36,37], HMGA2 [38], CEACAM1 [39,40], and mTOR [41] have been reported to be involved in proliferation and apoptosis in a number of cancers. There is a strong possibility that miR-581 regulates CRC proliferation and apoptosis by targeting these genes.

In conclusion, the current study demonstrates that miR-581/SMAD7 axis is crucial for CRC metastasis. Our results may provide a new insight into CRC molecular pathogenesis. MiR-581 might be a potential target in the therapy of CRC patients, especially with the metastasis form.

## 4. Materials and Methods

### 4.1. Affymetrix Microarray Data and Pre-Processing

We obtained miRNA and mRNA expression profiles of CRC fresh frozen tissues from GSE35834 in the Gene Expression Omnibus (GEO) database. The miRNA expression profile comprised 23 normal adjacent mucosa, 31 with primitive colorectal cancer, and 24 with liver metastasis. The corresponding number of the mRNA expression profile comprised 23 normal adjacent mucosa, 30 primitive colorectal cancer, and 27 liver metastases. We acquired microarray experiment results files andCEL files, and extracted miRNA and mRNA expression microarray datasets by performing affy [42] and oligo [43] R package, respectively. The Robust Multiarray Average (RMA) method [44] was used to normalize the data. By performing hierarchical cluster analysis in CRC samples, we identified and removed the outlier samples, and this process was accomplished by the *hclust* function in *WGCNA.*

### 4.2. STEM Analysis of MiRNA Expression Profile

We used the Short Time series Expression Miner [7] (http://www.cs.cmu.edu/~jernst/stem/) (STEM) algorithm and software to identify the variation of miRNA expression profiles in normal, tumor, and metastasis CRC samples. STEM Clustering Method was selected, and the parameter of minimum absolute expression change was set as 0.585. Other parameters were set to default values. MiRNAs were clustered according to the expression variation and the statistically significant modules were colored.

### 4.3. Weighted Gene Coexpression Network Analysis

In order to reveal the correlation of genes and to investigate the significantly correlated gene modules between normal, tumor, and metastasis CRC patients, we performed co-expression network analysis by using *WGCNA* R (https://www.r-project.org/) package. The soft thresholding power for network construction was determined through the *pickSoftThreshold* function. We constructed a weighted gene co-expression network based on the normal, tumor, and metastasis CRC expression data. Firstly, we created a matrix of adjacencies by using the adjacency function of *WGCNA*, and determined the concordances of gene expression by Pearson correlation values between gene pairs. Next, we transformed the matrix into a Topological Overlap Matrix (TOM) using *TOMsimilarity* function. Then, all genes were hierarchically clustered based on TOM matrix and generated a cluster dendrogram, which was used to determine modules by dynamic tree cut method.This analysis was performed by *cutreeDynamic* function of *WGCNA* (minModuleSize = 30), and we named modules with different colors. Finally, we calculated the correlation between clinical traits and modules, and selected the module that had significant correlation with CRC metastasis for further study.

### 4.4. Cell Line and Culture

Human colorectal cancer cell line SW480 was cultured in RPMI-1640 medium (Gibco, Gaithersburg, MD, USA) supplemented with 10% fetal bovine serum (Gibco, Gaithersburg, MD, USA). The cells were maintained in a humidified atmosphere at 37 °C with 5% CO_2_. 

### 4.5. Transfection and Reagents

We chemically synthesized miR-581 mimics, miR-581 inhibitor, and SMAD7 siRNA (target sequence: AGGCAUUCCUCGGAAGUCA). Before transfection assays, cells were inoculated into 6-well plates at 40–50% confluence. The transfection was performed using HiPerFect transfection reagent (Qiagen, Waltham, MA, USA) in accordance with the manufacturer’s instructions [6].

### 4.6. Dual Luciferase Reporter Assay

The wild types of SMAD7-3′UTR, including the binding bite of miR-581 and the site-directed mutant type, were synthesized and inserted into the pmiR-GLO™ luciferase reporter plasmid (Promega, Madison, WI, USA). The WT, or MUT vector, was co-transfected with miR-581 mimics or negative controls in 24-well plates using Lipofectamine 2000 (Invitrogen, Carlsbad, CA, USA). After 24 h, the cells were harvested and assayed by a Dual-Luciferase Reporter Assay (Promega, Madison, WI, USA) according to the manuals.

### 4.7. Western Blot

Western blot assay was carried out as previously described [45]. Immunoblot was carried out with rabbit anti-SMAD7 (ImmunoWay, Plano, TX, USA; YN2330, 1:1000), mouse anti-E-cadherin (BD Biosciences, San Jose, CA, USA; 610181, 1:1000), mouse anti-N-cadherin (BD Biosciences, San Jose, CA, USA; 610920, 1:1000), rabbit anti-Vimentin (CST, Danvers, MA, USA; #5741, 1:1000), and mouse anti-GAPDH (ImmunoWay, Plano, TX, USA; YM3029, 1:5000). Proteins were visualized with HRP Substrate (Millipore Corporation, Temecula, CA, USA) and then detected using the Chemiluminescence Imaging System (Tanon, Shanghai, China). 

### 4.8. Cell Proliferation Assay

Cell proliferation was measured by Cell Counting Kit-8 (CCK-8, 7Sea Pharmatech, Shanghai, China) assay. After 48 h of transfection, cells were seeded into 96-well plates at a density of 2 × 10^3^ cells/well with five replicates, and cultured for 1, 2, 3, and 4 days. Then, 10 μL CCK-8 solution and 90 μL of complete medium were added to each well and incubated for 3 h at 37 °C. The absorbance at a wavelength of 450 nm was measured to evaluate the number of viable cells.

### 4.9. Cell Invasion Assay

Cell invasion capacity was assessed using the transwell chamber (Costar, New York, NY, USA) coated with Matrigel (Corning, New York, NY, USA). Cells were starved and treated with mitomycin C (10 μg/mL, Sigma-Aldrich, Burlington, MA, USA) for 2 h before plating. A total of 5 × 10^4^ SW480 cells suspended in a 100 μL volume of serum-free medium were plated into the upper chamber, and 700 µL culture medium containing 10% FBS was added into the 24-well plate. After an incubation of 48 h, the cells in the upper chamber were carefully removed with cotton swabs, and the cells that passed through the membrane were fixed with 4% paraformaldehyde and stained with 0.1% crystal violet. Invaded cells were photographed under a microscope at a magnification of 100×. The cell number in five random fields was counted to evaluate the invasion ability. 

### 4.10. Statistical Analysis

All experiments were performed at least three times, and the data were presented as mean ± standard deviation. Statistical analysis was performed using GraphPad Prism v6 software (GraphPad Software Inc., La Jolla, CA, USA). A two-tailed *t*-test was used to assess the differences between groups. Differences were considered statistically significant when *p* < 0.05.

## Figures and Tables

**Figure 1 ijms-21-06499-f001:**
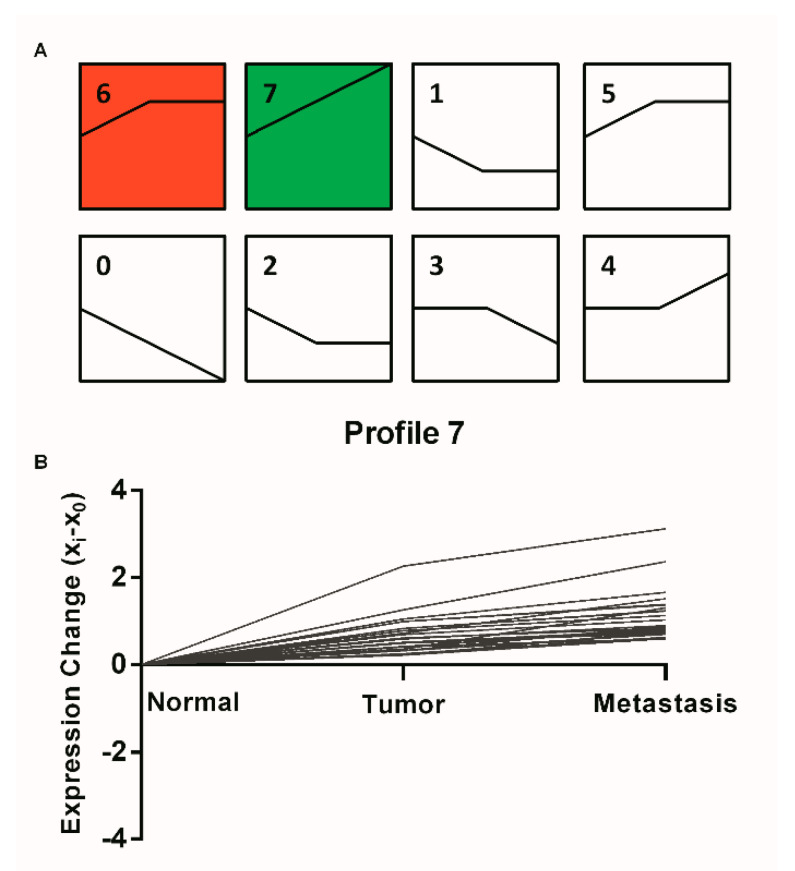
Short Time series Expression Miner (STEM) analysis identified the expression profiles of mRNAs. (**A**) Colored clusters indicated statistically significant numbers of genes enriched. The number in the top left represented model identifier. (**B**) In expression patterns of profile 7, each line in the figure represents an expression value of the corresponding miRNAs.

**Figure 2 ijms-21-06499-f002:**
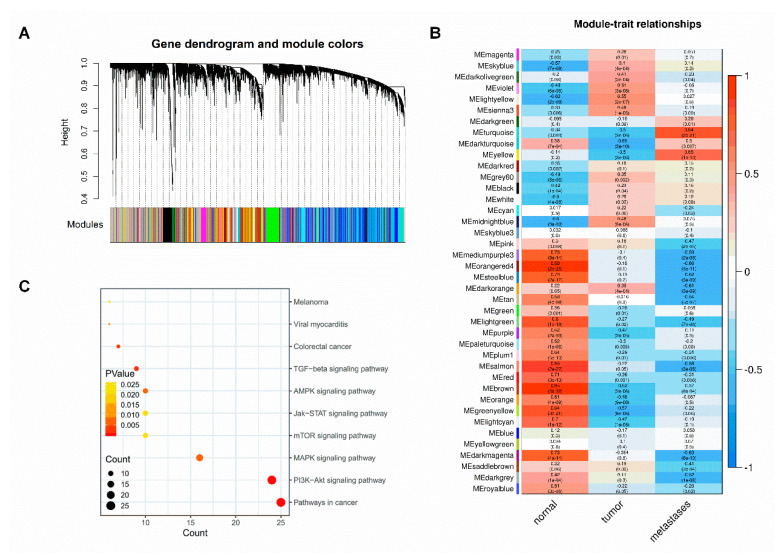
Identification of modules associated with clinical information, and pathways associated with metastasis. (**A**). Dendrogram of all expressed genes clustered based on a dissimilarity measure (1-TOM). The upper presents the clustering dendrogram of genes and the lower presents the assigned module colors. (**B**) Heatmap of module-trait correlations. Each cell describes the correlation coefficients and *p*-value. The cells are colored by the intensity of correlation, red for positive correlation and blue for negative correlation. The turquoise module was identified as trait-related modules for metastasis. (**C**). Kyoto Encyclopedia of Genes and Genomes (KEGG) enrichment analysis of genes in turquoise module. The depth of color corresponds to the enrichment significance, the size of the circle indicates the enriched gene count.

**Figure 3 ijms-21-06499-f003:**
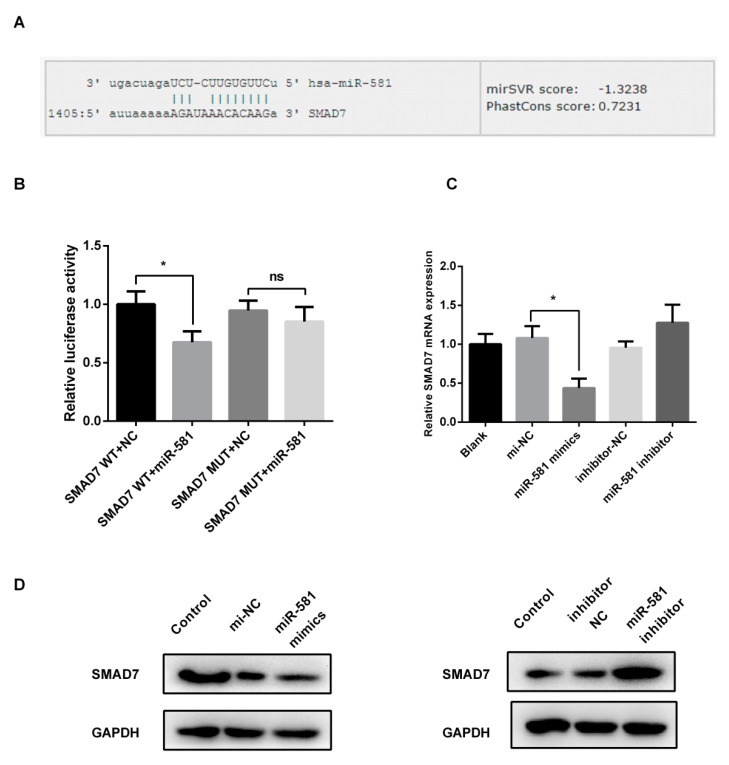
SMAD7 is a target of miR-581. (**A**) Binding site of miR-581 and SMAD7 3′-UTR, obtained from microRNA.org. (**B**) Results of the dual-luciferase reporter assay, and relative luciferase activity of each group was shown, * *p* < 0.05. (**C**) mRNA analysis of SMAD7 in miR-581 overexpression and down-regulation of cells. (**D**) Protein analysis of SMAD7 in miR-581 overexpression (left) or down-regulation (right) of cells.

**Figure 4 ijms-21-06499-f004:**
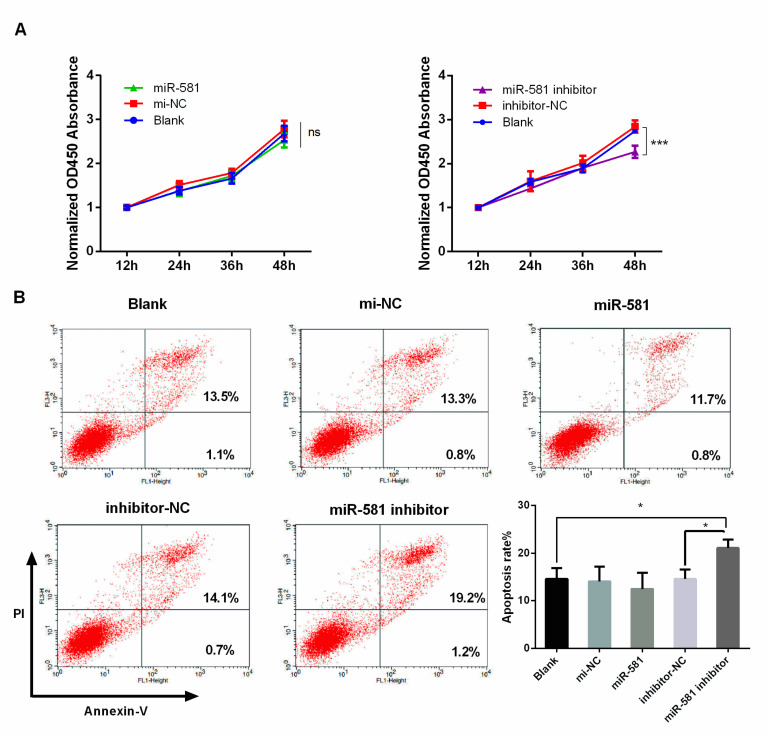
MiR-581 inhibition suppressed cell proliferation and induced apoptosis. (**A**). CCK-8 analysis of cell viability of SW480 cells overexpressing or knocking down miR-581. (**B**) Cell apoptosis was detected by flow cytometry. * *p* < 0.05 and *** *p* < 0.001.

**Figure 5 ijms-21-06499-f005:**
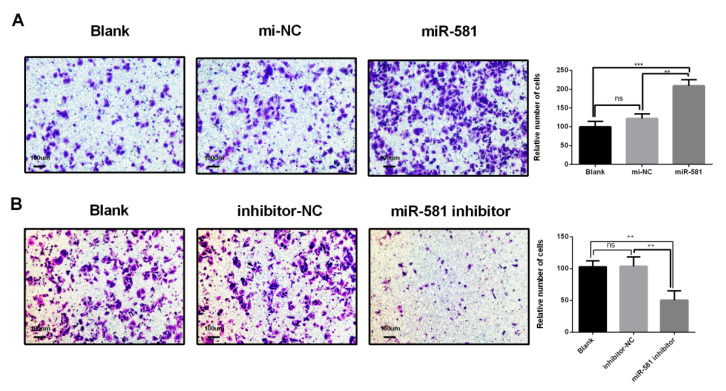
MiR-581 promotes cell invasion capacity. (**A**) MiR-581 overexpressing SW480 cell invasion capacity, as detected by transwell assay (100×). (**B**) MiR-581-depletion SW480 cell invasion capacity, as detected by transwell assay (100×). Relative number of invaded cells is shown in the histogram. ** *p* < 0.01 and *** *p* < 0.001.

**Figure 6 ijms-21-06499-f006:**
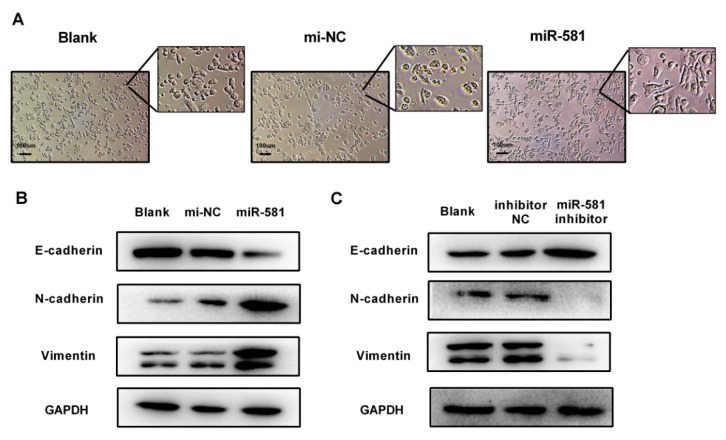
MiR-581 modulates the epithelial to mesenchymal transition (EMT) process in SW480 cells. (**A**) Cell morphological changes in epithelial SW480 cells after transfection of miR-581 mimics and negative control. (**B**) EMT markers E-cadherin, N-cadherin, and Vimentin were detected in miR-581-overexpressing cells. (**C**) EMT markers E-cadherin, N-cadherin, and Vimentin were detected in miR-581 knockdown cells.

**Figure 7 ijms-21-06499-f007:**
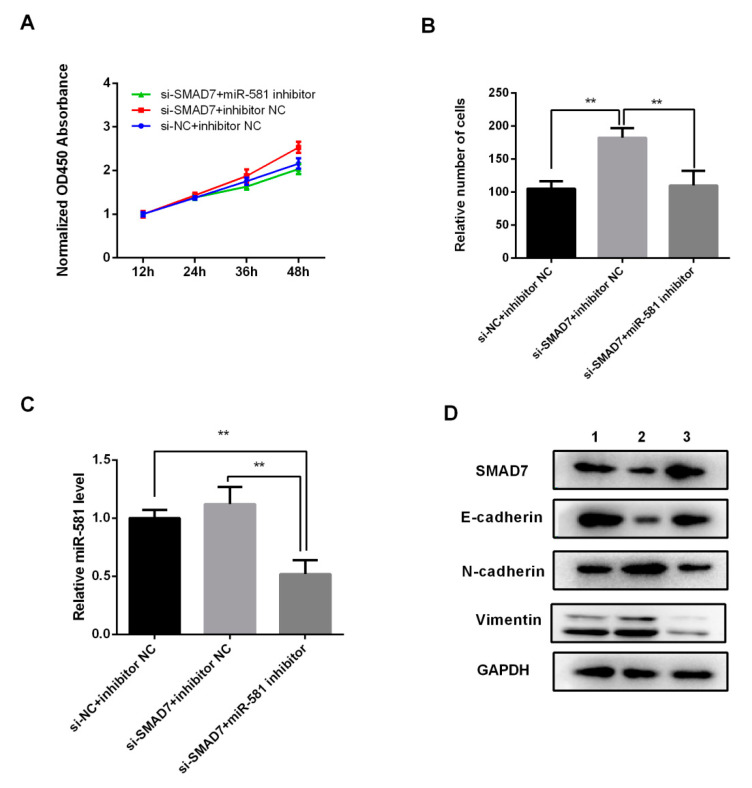
MiR-581 inhibition reversed the malignant phenotype induced by SMAD7 knockdown. (**A**) Cell proliferation was determined by CCK-8 assay. (**B**) Relative number of invaded cells in SMAD7 knockdown group, co=transfection group, and NC group. (**C**) miR-581 level was detected by qPCR. (**D**) EMT markers E-cadherin, N-cadherin, and Vimentin were detected by Western blot. Lane 1: si-NC + inhibitor NC; Lane 2: si-SMAD7 + inhibitor NC; Lane 3: si-SMAD7 + miR-581 inhibitor. ** *p* < 0.01.

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
