# Peer review of "MiR-581/SMAD7 Axis Contributes to Colorectal Cancer Metastasis: A Bioinformatic and Experimental Validation-Based Study"

_ijms, 2020, doi:10.3390/ijms21186499_

Round 1

Reviewer 1 Report

In the study "miR-581/SMAD7 axis contributes to colorectal cancer metastasis: a bioinformatic and experimental validation based study" the authors have done a remarkable job in compiling the data in a novel direction. However, there is a lot of grammatical mistakes. it seem the presentation quality of results and grammar has gone down than the previous version of manuscript itself and it needs to be heavily corrected. For example, giving you some line numbers which harbor the mistakes, the lines are as follow, 2,16,21,30,34,62,71,112 and so on. Second big thing is, the way of high throughput data presentation, which is still confusing and raw. I would suggest the authors to look up some papers where this kind of work has been done, in order to better present the data.      

Author Response

Point 1: However, there is a lot of grammatical mistakes. it seem the presentation quality of results and grammar has gone down than the previous version of manuscript itself and it needs to be heavily corrected. For example, giving you some line numbers which harbor the mistakes, the lines are as follow, 2,16,21,30,34,62,71,112 and so on.

Response 1: These mistakes have been revised, and we have carefully revised the manuscript and corrected the grammatical mistakes, as shown in the manuscript.

Point 2: Second big thing is, the way of high throughput data presentation, which is still confusing and raw. I would suggest the authors to look up some papers where this kind of work has been done, in order to better present the data.

Response 2: We had look up a lot of number papers about high throughput data analysis when we wrote the manuscript. As shown in “Materials and Methods” section, the sample raw files could be acquired in Gene Expression Omnibus (GEO) database (a public database), the number of sample and sample numbers that we selected in this study were also presented in supplement figures.

Bioinformatic analysis methods can be viewed on the official website, and we also annotate the corresponding references in the manuscript. The key parameters that need to be set are listed in “Materials and Methods”. I believe readers in bioinformatics field can repeat the results according to the information.  In addition, the original analysis data is too large and cannot be shown graphically, but the crucial results were shown in figures and supplemental tables. It can guide readers to repeat the results.

Reviewer 2 Report

This a very interesting study based on bioinformatics. To readers not in the field you should explain the term "Turquoise module" in the methods or the first time you mention it, I got a little lost here. Otherwise it was logical and the figures were appropriate. One figure, Fig 2 needs to be looked at as I'm sure that in the legend B and C are reversed. Please address this issue.

Other required corrections are marked in the pdf attached

Author Response

Point 1: To readers not in the field you should explain the term "Turquoise module" in the methods or the first time you mention it, I got a little lost here. Otherwise it was logical and the figures were appropriate. 

Response 1: Actually, we name modules with different colours, "Turquoise module" is the module marked Turquoise colour in Figure 2B. we have perfected the manner in Materials and Methods in the manuscript.

Point 2: One figure, Fig 2 needs to be looked at as I'm sure that in the legend B and C are reversed. Please address this issue.

Response 2: This mistake has been revised, we exchange figure 2B with 2C in picture.

Point 3: Other required corrections are marked in the pdf attached

Response 3: We have carefully revised the manuscript and corrected the corrections.

Round 2

Reviewer 1 Report

In the study "MiR-581/SMAD7 axis contributes to colorectal cancer
metastasis: a bioinformatic and experimental validation based study" the authors have done good job and answered all my concerns. I congratulate them for their good work.